# Federated Learning Backdoor Attack Based on Frequency Domain Injection

**DOI:** 10.3390/e26020164

**Published:** 2024-02-14

**Authors:** Jiawang Liu, Changgen Peng, Weijie Tan, Chenghui Shi

**Affiliations:** 1State Key Laboratory of Public Big Data, College of Compute Science and Technology, Guizhou University, Guiyang 550025, China; 15863162557@163.com (J.L.); wjtan@gzu.edu.cn (W.T.); 2Key Laboratory of Advanced Manufacturing Technology of Ministry of Education, Guizhou University, Guiyang 550025, China; 3College of Computer Science and Technology, Zhejiang University, Hangzhou 310058, China; chenghuishi@zju.edu.cn

**Keywords:** federated learning, backdoor attack, frequency domain, Fourier transform

## Abstract

Federated learning (FL) is a distributed machine learning framework that enables scattered participants to collaboratively train machine learning models without revealing information to other participants. Due to its distributed nature, FL is susceptible to being manipulated by malicious clients. These malicious clients can launch backdoor attacks by contaminating local data or tampering with local model gradients, thereby damaging the global model. However, existing backdoor attacks in distributed scenarios have several vulnerabilities. For example, (1) the triggers in distributed backdoor attacks are mostly visible and easily perceivable by humans; (2) these triggers are mostly applied in the spatial domain, inevitably corrupting the semantic information of the contaminated pixels. To address these issues, this paper introduces a frequency-domain injection-based backdoor attack in FL. Specifically, by performing a Fourier transform, the trigger and the clean image are linearly mixed in the frequency domain, injecting the low-frequency information of the trigger into the clean image while preserving its semantic information. Experiments on multiple image classification datasets demonstrate that the attack method proposed in this paper is stealthier and more effective in FL scenarios compared to existing attack methods.

## 1. Introduction

The development of big data has promoted the widespread application of artificial intelligence technology. The performance of deep learning models heavily relies on the quantity and quality of training data, and reasons such as industry competition, legal requirements for data privacy, intellectual property protection, and data silos have emerged [1]. For example, in an organization, departments have their own data. These data are related to each other but exist independently in different departments. With respect to the concerns of security, privacy, and other aspects, each department can only obtain the data of its own department and cannot obtain data from other departments. It is like the sea of information technology, and data are stored and defined separately, forming isolated islands in the sea, that is, “data silos”. The proposition of federated learning (FL) [2] aims to address this challenge by enabling parties to train models without sharing their data locally. Owing to its privacy-preserving nature, FL has found extensive applications in numerous data-sensitive domains, such as finance [3], healthcare [4], and security [5]. Thus, FL represents one of the most promising paradigms in privacy-preserving distributed learning nowadays.

However, the privacy-preserving features of FL also provide conveniences for attackers, among which backdoor attacks are a common threat in federated learning [6,7]. During the model training process, related information about the model (such as the model parameters, architecture, and gradient parameters) can be exchanged among participants, but the local data will not leave the local area. Beyond this inherent privacy protection mechanism, the practical implementation of FL systems further utilizes Secure Multi-party Computation (MPC) [8] techniques to protect each client’s intermediate computational results. The model parameters uploaded by each client are invisible not only to other participants but also to the server. Ironically, attackers can deploy nearly any attack payload under the protection of the FL protocol itself using manipulated clients.

A backdoor attack is a targeted attack that involves the intentional introduction of harmful data or the manipulation of data during the training process in order to be able to activate specific backdoors once the model training is complete. These backdoors cause the model to exhibit abnormal or predetermined behavior when it encounters data with triggers, In FL, attackers inject backdoors into the global model by polluting the training data sets of participants or directly manipulating malicious clients to submit malicious model updates to the server [9]. Backdoor attacks pose a serious security threat in classification tasks such as autonomous driving [10], medical analysis [11], or scene classification [12]. Consider a traffic recognition task in an autonomous vehicle. A model with a misleading backdoor will lead to a misjudgment of the STOP sign as a growth limit sign. In the field of medical analysis, models with backdoors implanted may mislead medical analysis, leading to a wrong diagnosis or prediction, thus having a significant impact on the health of patients.

Backdoor attacks in FL have been studied in many papers. The most common trigger of backdoor attacks in federated learning is a pixel pattern [9,13,14]. However, pixel-pattern triggers exhibit several shortcomings. First, their stealthiness is not good, as they are easily detectable by human eyes. Second, pixel-pattern triggers, when applied in the spatial domain, alter the spatial pixel information. This leads to a discrepancy between the poisoned sample’s label and its semantic representation, manifested as incorrectly annotated instances. Such inconsistencies significantly diminish the stealthiness of the attacks. These flaws lead to the existing FL backdoor attacks.

To solve the above problems, we propose a novel FL backdoor attack based on frequency-domain injection. First, we perform a Fourier transformation on the clean image and the trigger image to obtain the amplitude and phase spectra of the two images. Second, the phase spectrum of the benign image is kept unchanged, while the spectral amplitudes of the two images are linearly mixed to synthesize a new spectral amplitude. Finally, the inverse Fourier transform is applied to the synthesized spectrum and the original phase spectrum to obtain the poisoned image. Since the amplitude spectrum can capture low-level distribution, the phase spectrum can capture high-level semantic information. The injected trigger amplitude spectrum does not change the spatial domain and retains the semantic information of the contaminated pixels. This achieves better attack stealthiness.

The contributions are summarized as follows:(1)We introduced a frequency-domain injection method, which significantly enhances the stealthiness of the trigger compared to the pixel-pattern trigger.(2)Multiple task scenarios are considered, and extended experiments in these task scenarios demonstrate the effectiveness and stealthiness of our proposed method.(3)By examining various defense strategies, it is demonstrated that these current defense strategies fail to detect our proposed attacks.

## 2. Related Work

Backdoor attacks: In centralized settings, current backdoor attacks primarily consider two approaches: (1) dirty-label attacks, which modify training samples and set their corresponding labels to the target label, and (2) clean-label attacks, which do not replace the original labels. In dirty-label attacks, Gu et al. [15] were among the first to study backdoor attacks in deep learning and to introduce BadNets, which inject triggers into a small randomly selected subset of the training set and further label them as the target category. Chen et al. [16] designed a backdoor attack based on image blending, where the trigger is designed as an additional image or random noise. Turner et al. [17] proposed a less conspicuous method of backdoor attacks, constructing triggers through adversarial perturbations without changing the image’s label. Luo et al. [18] developed triggers for each image using a generator without changing the image labels. Additionally, some studies on clean-label attacks have attempted to perturb inputs of the target category so that the perturbed samples can mimic backdoor inputs from non-target categories.

Backdoor defenses: In centralized scenarios, defenses can be divided into during-training and post-training categories. During the training process, defenders can detect poisoned samples or invalidate the poisoning process by considering poisoned data as outliers. This can be completed using robust statistical methods in the input space or techniques in the feature space to detect and eliminate these poisoned samples. However, these defenses may reduce the model’s performance or accuracy, particularly by generating more errors in normal data. This could make them unsuitable for distributed environments. Post-training defenses, such as neural cleanse [19] and fine-tuning [20], are applied to models that have already been trained and can be used in distributed settings. They work by identifying and mitigating the effects of backdoor attacks, thus making the models shared in a distributed learning environment more robust and reliable.

In FL, to mitigate the effects of backdoor attacks prior to aggregation, numerous secure aggregation algorithms have been proposed [20,21,22]. At the point of client-to-server aggregation, there is a discernible difference between the vector spaces of malicious and benign clients. These methods initially identify malicious clients as outliers in the distribution of local model updates, subsequently excluding them from aggregation. However, these methods are only effective under specific attacks and are based on detailed assumptions about the attacks or data distributions. They are primarily targeted at Byzantine attacks and are not applicable in backdoor attack scenarios. Several studies have also focused on differential privacy approaches. For instance, Weak-DP [9] mitigates backdoor attacks by clipping the norm of the global model and adding Gaussian noise. CRFL [23] employs clipping and smoothing of model parameters, generating a sample-based robustness certification, where the size of the backdoor trigger pattern is restricted. Ozdayi et al. [24] attempt to enhance the robustness of FL by assigning different learning rates to each client.

## 3. Threat Model

### 3.1. Federated Learning Process

Assuming the existence of C clients, with each client possessing a dataset of size $n_{i}$, denoted as $D_{i}$, the collective dataset size across all clients amounts to $N=\sum_{i=1}^{n}n_{i}$. During the t-th round of FL training, the server selects a subset of m clients from the set of C clients and sends the aggregated model $\theta_{t}$. Following that, the client receives the aggregated model $\theta_{t}$ and conducts local training for K rounds, resulting in the model $\theta_{t}^{i,k}$. Then, the client sends the updates $\theta_{t}^{i,k}-\theta_{t}$ to the server. Now, on the server side, aggregation of updates received from clients is performed to obtain the new aggregated model $\theta_{t+1}$ for the next round. In the standard federated learning averaging algorithm, the server receives the weighted average of updates from m clients, where the weights are typically determined by the number of samples or other criteria:(1)$$\theta_{t+1}=\theta_{t}+\frac{1}{N}\sum_{i=1}^{m}n_{i}\left(\theta_{i}^{i,K}-\theta_{t}\right)$$

### 3.2. Attacker Capabilities

Attackers can manipulate the training data of malicious clients and intervene in the hyperparameters of local clients, such as the number of training iterations and the learning rate. Prior to aggregation with the server, attackers are able to modify the model’s weight parameters. Furthermore, attackers can adaptively alter the local training process.

### 3.3. Attacker Objectives

Our attacker aims to create a joint model through FL, achieving high accuracy on both its primary task and the backdoor subtask selected by the attacker, while maintaining this high accuracy on the backdoor subtask across multiple rounds post-attack. Additionally, it is essential to ensure that the current local model being trained does not deviate excessively from the global model. The stealthiness of the attack is reflected in the fact that the addition of the trigger does not cause significant appearance differences in the image data, enabling it to withstand defenses.

## 4. Method

In this section, we introduce a federated learning backdoor attack based on frequency-domain injection. Firstly, we provide annotations for the symbols used in the paper.

**Table d66e494:** 

Notations	Meanings
F(x)	Fourier transform function
$F^{-1}\left(x\right)$	Inverse Fourier transform function
$A_{x_{i}}$	Image amplitude spectrum
$P_{x_{i}}$	Image phase spectrum
$x_{i}$	Clean image
$x^{t}$	Trigger image
$D_{train}$	Training set
α	The mixing ratio of $A_{x_{i}}$ and $A_{x^{t}}$
β	Low-frequency plaque range
*M*	Binary mask matrix
$L_{\;cln}\;and\;L_{mal}$	Cross entropy function
$x_{i}^{p}$	Poisoned image
$L_{m}^{t+1}$	Malicious client updates
ε	Model update threshold
$\theta_{G}^{t}$	Global model aggregation after round *t*
$\theta_{L}^{t+1}$	Latest local model of client C after round *t* + 1

Our method is divided into two stages, such as Algorithm 1. In the first stage, a frequency-domain transformation is used to construct invisible poisoning samples and realize the trigger stealthiness. In the second stage, the effectiveness of the attack is achieved by expanding the weight update of the malicious client, and the Projected Gradient Descent (PGD) method is introduced on the server side to constrain the local model update to achieve the stealth of the attack.
**Algorithm 1** Federated learning backdoor attack based on frequency injection**Input**: Benign dataset $D_{train}^{b}$, benign image $x_{b}$, trigger image $x^{t}$, local batch size B, the number of the local training round R, the global model after the t th round of aggregation $\theta_{G}^{t}$, the latest local model of client C after round t + 1 $\theta_{L_{i}}^{t+1}$.**Output:** Malicious client model update $\theta_{L_{i}}^{t+1}$.**Stage 1**: Create a poisoned image.1.**For** all $\left(x_{i},y_{i}\right)\in D_{train}^{b}$, complete the following:2.Perform Fast Fourier Transform of $x^{t}$ and $x_{i}$// Perform Equation (4) to perform a fast Fourier transform.3.$A_{x_{i}}^{P}=\left[\left(1-\alpha \right)A_{x_{i}}+\alpha A_{x^{t}}\right]\cdot M+A_{x_{i}}\left(1-M\right)$// Equation (5) is executed to obtain $A_{x_{i}}$ and $A_{x^{t}}$, and the amplitude spectra of the trigger image $A_{x^{t}}$ and the clean image $A_{x_{i}}$ are mixed by a binary mask matrix M.4.$x_{i}^{P}=F^{-1}\left(A_{x_{i}}^{P},P_{x_{i}}\right)$//$P_{x_{i}}$ is obtained through Equation (5), and the original phase $P_{x_{i}}$ and amplitude spectra $A_{x_{i}}^{P}$ are synthesized for the inverse Fourier transform to obtain $x_{i}^{P}$.5.Add $x_{i}^{p}$ to $D_{train}^{b}$ to obtain $D_{train}^{p}$.6.**end for**7.**Return** to the poisoned training set $D_{train}^{p}$.**Stage 2**: Federated learning backdoor attack.8.The server sends global model parameters $\theta_{G}^{t}$ to the client and updates the local model.9.**For** R = 1,…R, complete the following:10.$B_{1}\leftarrow$(split $D_{train}^{p}$ into batches of size B).11.**For** $b_{1}\in B_{1}$, complete the following:12.$\theta_{L_{i}}^{t+1}=\theta_{L_{i}}^{t+1}-\eta \nabla L_{class-loss}$// Perform stochastic gradient descent algorithm to update the local model.13.**If** $\left\|\theta_{L_{i}}^{t+1}-\theta_{G}^{t}>\varepsilon \right\|$, complete the following: // Execute the PGD algorithm to constrain the local model update magnitude not to exceed a given threshold value.14.$\theta_{L_{i}}^{t+1}=\theta_{G}^{t}+\frac{\left(\theta_{L_{i}}^{t+1}-\theta_{G}^{t}\right)}{{\left\|\theta_{L_{i}}^{t+1}-\theta_{G}^{t}\right\|}_{2}}\times \varepsilon$//${\left\|\theta_{L_{i}}^{t+1}-\theta_{G}^{t}\right\|}_{2}$ denotes the L2 norm of the update weights.15.**end for**16.**end for**17.**Return** $\theta_{L_{i}}^{t+1}$ to server.

### 4.1. Frequency Domain Poisoned Sample Generation

Currently, most trigger generation methods in backdoor attacks involve altering pixels in the spatial domain to create poisoned samples, but changing pixel information affects the spatial layout of the image and is easily perceived by the human eye. By implanting triggers in the frequency domain, better stealth can be achieved. This is because, after performing a Fourier transform on an image, we obtain its amplitude and phase spectra. The amplitude spectrum captures low-level distributions, while the phase spectrum encodes high-level semantic information. Changes in the amplitude spectrum do not significantly affect the perception of high-level semantics. Therefore, we can linearly mix the amplitude spectra of two images to synthesize a new amplitude spectrum, preserving the spatial semantic information and enhancing stealthiness. As shown in Figure 1, the process is as follows:

For a clean sample $x_{c}\in D_{train}$ and a trigger image $x_{t}$, both undergo a Fast Fourier Transform (FFT). That is,
(2)$$F\left(x_{i}\right)\left(u,v,c\right)=\overset{H-1}{\underset{h=0}{\sum }}\overset{W-1}{\underset{w=0}{\sum }}x_{i}\left(h,w,c\right)e^{-j2\pi \left(\frac{h}{H}\mu +\frac{w}{W}v\right)}$$

The amplitude and phase spectra of $x_{i}$ and $x^{t}$ are defined as follows:(3)$$\{\begin{array}{c} A_{x_{i}}=F^{A}\left(x_{i}\right),A_{x^{t}}=F^{A}\left(x^{t}\right)\\ P_{x_{i}}=F^{P}\left(x_{i}\right),P_{x^{t}}=F^{P}\left(x^{t}\right) \end{array}$$

Subsequently, by blending the amplitude spectra of the trigger images $A_{x^{t}}$ and $A_{x_{i}}$, you obtain $A_{x_{i}}^{P}$, finally introducing a binary mask, $M=1_{\left(h,w\right)\in \left[-\beta H:\beta H,-\beta W:\beta W\right]}$, where β determines the position and range of low-frequency patches within the amplitude spectrum, with a value of 1 inside the amplitude spectrum and 0 elsewhere. α represents the mixing ratio of information from $A_{x_{i}}$ and $A_{x^{t}}.$ Therefore, the composite amplitude spectrum of the final synthesized image is expressed as follows:(4)$$A_{x_{i}}^{P}=\left[\left(1-\alpha \right)A_{x_{i}}+\alpha A_{x^{t}}\right]\cdot M+A_{x_{i}}\left(1-M\right)$$

Finally, the poisoned image is generated by combining the composite amplitude spectrum with the original phase spectrum $P_{x_{i}}$:(5)$$x_{i}^{p}=F^{-1}\left(A_{x_{i}}^{P},P_{x_{i}}\right)$$

Therefore, by linearly blending the spectral amplitudes of two images in the frequency domain, a new spectral amplitude is synthesized, which preserves spatial semantic information and achieves improved stealthiness. Then, the inverse FFT is applied to the synthesized spectrum of the benign image and the original phase spectrum to generate the poisoned image.

### 4.2. Model Backdoor Injection and Submission

As shown in Figure 2, the attacker adds the poisoned samples generated by the frequency-domain-based injection to local client 2 and implants a backdoor to the global model by augmenting the parameters of the malicious client. When the global model with the backdoor is tested, the photo with the trigger dog is recognized as a cat.

The attacker selects one or more local clients to attack and adds the poisoned image $x_{i}^{p}$ to the malicious client training set. Suppose there are n clients in the federated learning system, denoted as $C=\left\{C_{1},C_{2},\ldots ,C_{n}\right\}$. For malicious clients, there are both clean data and poisoned data; when malicious clients are trained, the correct classification accuracy is guaranteed on clean samples, and the wrong classification is guaranteed on poisoned samples. During the training, the loss of cross-entropy becomes smaller and smaller due to the fact that no labels are changed on the clean samples, and the model’s prediction results and the real labels become closer and closer to the real labels, which guarantees the correct classification on the clean samples. On the poisoned samples, since the backdoor attack is a directed poisoning attack, the attacker specifies the label τ of the attack; e.g., the label of the dog of those poisoned samples is specified as the picture of the cat to be attacked. Making the model’s prediction as close as possible to the label specified by the attacker ensures that the poisoning samples are misclassified. These loss functions are denoted as $L_{class-loss}$, and they are defined as follows:(6)$$L_{class-loss}=L_{cln}+L_{mal}$$

Bringing in the cross-entropy loss function yields, the following is obtained:(7)$$L_{class-loss}=-\overset{K}{\underset{i=1}{\sum }}y_{i}\log\left(p_{i}\right)+\left(-\overset{K}{\underset{i=1}{\sum }}\tau \log\left(p_{i}\right)\right)$$
where $p_{i}$ is the probability of the predicted category of the local model, K is the number of labeled categories in the dataset, $y_{i}$ is the true label, and τ is the label specified by the attacker.

If the attacker executes the Stochastic Gradient Descent (SGD) algorithm for too long, then the generated model can deviate severely from its origin, thus making a simple paradigm tailoring defense effective. Therefore, in order to improve the stealthiness of backdoor attacks, we further propose to use PGD to constrain the update of the local model from exceeding a given threshold ϵ during backdoor injection by the local client:(8)$$\left|\theta_{L_{i}}^{t+1}-\theta_{G}^{t}\right|<\varepsilon$$

The adversary then runs PGD where the projection happens on the ball centered around $\theta_{L_{i}}^{t+1}$ with radius ε.

In the context of backdoor attacks on federated learning, it is essential to consider the effectiveness of such attacks. Firstly, the weights of malicious clients are likely to be diminished by the aggregation algorithms used on the server side. Secondly, during the training process of FL, there is no guarantee that malicious clients will be selected in every round. Inspired by the work of Bagdasaryan and others [21], the malicious clients have already accounted for the performance on both the normal dataset and the tampered, poisoned dataset within their loss function. Assuming this model is referred to as Model *X*, the ideal outcome after aggregation would be the result equivalent to Model *X*:(9)$$X=\theta_{G}^{t}+\frac{\eta }{n}\left(\theta_{L_{i}}^{t+1}-\theta_{G}^{t}\right)$$

For the normal client $C_{i},i=1,\ldots ,m-1$, as the model approaches convergence, the equation is as follows:(10)$$\overset{\;m}{\underset{i=1}{\sum }}\left(\theta_{L_{i}}^{t+1}-\theta_{G}^{t}\right)\approx 0$$

Thus, the local model submitted by the malicious client $C_{m}$ satisfies the condition that
(11)$$\;L_{m}^{t+1}=\frac{n}{\eta }X-\left(\frac{n}{\eta }-1\right)\theta_{G}^{t}-\overset{m-1}{\underset{i=1}{\sum }}\left(\theta_{L_{i}}^{\theta +1}-\theta_{G}^{t}\right)$$

Setting $\lambda =\frac{n}{\eta }$ and simplifying, we obtain the following:(12)$$L_{m}^{t+1}\approx \lambda \left(X-\theta_{G}^{t}\right)+\theta_{G}^{t}$$

## 5. Experimental Setting

### 5.1. Dataset and Models

We used three image classification tasks: CIFAR-10, GTSRB, and ISIC-2019. The CIFAR-10 dataset consists of 10 classes, with each image having a size of 32 × 32. There are 6000 images per class, making a total of 50,000 training images and 10,000 test images in the dataset. The GTSRB dataset is used for traffic sign recognition and contains 43 classes of traffic signs. It consists of 39,209 training images and 12,630 test images. The ISIC-2019 dataset includes 25,331 skin disease images belonging to 8 diagnostic categories, including melanoma, nevus, basal cell carcinoma, actinic keratosis, benign keratosis, dermatofibroma, vascular lesion, and squamous cell carcinoma. For the classification tasks on these three datasets, we employed the ResNet-18 as the base model. The ResNet-18 architecture consists of one convolutional layer with a 3 × 3 kernel and a stride of 1, four BasicBlocks, and one fully connected layer. Each BasicBlock contains two convolutional layers with 3 × 3 kernels. The stride for the two convolutional layers in the first BasicBlock is 1, while the other BasicBlocks have strides of 2 and 1. The output channels for each BasicBlock are 64, 128, 256, and 512, respectively.

### 5.2. FL Parameters

In FL, there are a total of 100 clients, and the dataset distribution considers both non-IID (non-independent and identically distributed) and IID (independent and identically distributed) settings. In the non-IID setting, the Dirichlet sampling parameter is set to the default value of 0.9. Each client conducts 2 rounds of training on their local data. The server’s learning rate is set to 0.01, and the SGD optimizer is used with a batch size of 64. The initial learning rate is set to 0.1 and is reduced by a factor of 5 every 100 training epochs. A total of 300 epochs are trained. Additionally, a scaling factor λ is set to 10. During each training round, 10 clients are selected for training.

### 5.3. Attack Method Parameters

In the context of image classification tasks, the values of α were set to 0.15, while the value of β was set to 0.2. In these classification tasks, the target labels for both training and testing were “horse”, “speed limit (70 km/h)”, and “melanocytic nevus”.

### 5.4. Evaluation Indicators

ASR: The Attack Success Rate is the rate of backdoor samples that are successfully classified as target labels. The ASR is used to measure the recognition accuracy of the backdoor model for backdoor data, and accuracy refers to the ability to accurately recognize the backdoor image as the target label, rather than its true label. The closer the ASR is to 1, the stronger the attack.

ACC: The model’s classification success rate on clean samples, i.e., the ratio of the number of samples correctly predicted by the model to the total number of samples. The ACC is used to measure the recognition accuracy of the backdoor model on clean data, where recognition accuracy refers to the ability to accurately recognize clean samples as true labels. The closer it is to 1, the better performance of the model.

PSNR (the Peak Signal-to-Noise Ratio): As the name suggests, the PSNR measures the pixel error corresponding to the addition of the frequency-domain trigger poisoning image I(i,j) and the original clean image K(i,j). A larger PSNR implies less distortion in the generated poisoned image.
(13)$$MSE=\frac{1}{mn}\sum_{i=0}^{m-1}\sum_{j=0}^{n-1}{\left[I\left(i,j\right)-K\left(i,j\right)\right]}^{2}$$

PSNR is defined as follows:(14)$$PSNR=10\cdot log_{10}\left(\frac{MAX_{I}^{2}}{MSE}\right)$$
where $MAX_{I}^{2}$ indicates the maximum value of the image color.

SSIM (the Structural Similarity Index): SSIM is a metric used to measure the similarity of two images. It is calculated based on the brightness and contrast of local patterns. For SSIM, the value ranges from 0 to 1. The higher the finger of the similarity index (almost close to 1), the closer the poisoned image is to the original clean image. It is defined as follows:(15)$$SSIM=\frac{\left(2\mu_{x_{c}}\mu_{x_{p}}+c_{1}\right)\left(2\sigma_{x_{c}x_{p}}C_{2}\right)}{\left(\mu_{x_{c}}^{2}+\mu_{x_{p}}^{2}+C_{1}\right)\left(\sigma_{x_{c}}^{2}+\sigma_{x_{p}}^{2}+C_{2}\right)}$$
where $x_{c}$ and $x_{p}$ are poisoned and clean samples, respectively, $\mu_{x_{c}}$ and $\mu_{x_{p}}$ are sample pixel mean values, and $\mu_{x_{c}}^{2}$ and $\mu_{x_{p}}^{2}$ are sample pixel variance values. $\sigma_{x_{c}x_{p}}$ is the sample pixel covariance, the constants are to maintain the stability, $C_{1}={\left(0.01L\right)}^{2}$, and $C_{2}={\left(0.03L\right)}^{2}$ is the pixel value dynamic range. L is the dynamic range of the pixel values.

## 6. Experiments

### 6.1. Backdoor Attack Effectiveness

In this section, we conduct experiments on the effectiveness of federated learning backdoor attacks. The ASR and ACC on two models for three datasets are evaluated, as well as the effectiveness of the attack in the One-shot Attack, Continuous attack, and Multiple Trigger backdoor attack scenarios.

#### 6.1.1. Main Results

From Table 1, it can be observed that without any attacks, the accuracy (ACC) on both ResNet18 and VGG13 models exceeds 70% on the CIFAR-10 and ISIC-2019 datasets and exceeds 90% on the GTSRB dataset. When the data is either IID or non-IID, the ACC of all three attack methods on the CIFAR-10 and ISIC-2019 datasets remains above 70%, and on the GTSRB dataset, it remains above 90%. Notably, when the data is IID, the ACC is slightly higher than when the data is non-IID.

Our proposed method shows similar attack effectiveness to the other two attack methods. Regardless of whether the data is IID or non-IID, our method’s attack success rate exceeds 99%. The effectiveness of the Blend attack is slightly lower than that of the Frequency and Pixel-pattern attacks, but it is still significant.

#### 6.1.2. One-Shot Attack and Continuous Attack

Single Attack: The attacker conducts only one attack, but in this round of the attack, the amplification factor is set to λ = 100, with the expectation of injecting a backdoor in a single attempt.

As shown in Figure 3 and Figure 4, the proposed method demonstrated greater durability than the blend trigger but was less effective than the pixel trigger. This could be attributed to the pixel trigger altering pixel information in the spatial domain, thereby preserving pixel semantics, which are less likely to be forgotten during model training. In contrast, the blend trigger disrupts a significant number of the semantic features in the spatial domain, leading to inferior attack effectiveness. Our approach involves synthesizing a new spectral amplitude by linearly blending the spectral amplitudes of two images in the frequency domain. This preserves spatial semantic information, making it easier for the model to remember.

In multiple attack scenarios, our method required a higher number of attacks to achieve better effectiveness. This is primarily due to the reduced modification extent in model updates, intended to enhance the stealthiness of the attack.

Continuous Attack: The attacker carries out uninterrupted attacks, but in this round of the attack, the amplification factor is set to λ = 1 in order to stealthily inject the backdoor.

#### 6.1.3. Multiple Trigger Backdoor Attack

We also evaluated the feasibility of simultaneously injecting multiple backdoors into the model. The training input for each backdoor is included in each batch of the attacker’s training data. The training ceases when the model converges on all backdoors (with each backdoor task achieving an accuracy of 95%). The more backdoors there are, the longer it takes for the model to converge.

The experimental results, as illustrated in Figure 5, show that the efficacy of multi-backdoor attacks is similar to that of a single backdoor in a single attack. After replacement, the global model immediately achieves at least 90% accuracy on all backdoor tasks. The main task accuracy decreases by less than 1%. Furthermore, the figure also displays the L2 norm of the models submitted by attackers under varying numbers of backdoors. As the number of backdoors increases, the magnitude of the L2 norm of the model updates submitted by attackers also increases (i.e., they become more easily detectable by the server).

### 6.2. Backdoor Attack Stealthiness

In this section, we conducted experiments on the stealthiness of federated learning backdoor attacks. The stealthiness of the poisoned images was measured by visualizing the residual images and calculating the PSNR and SSIM metrics. The stealthiness and effectiveness of our method are also evaluated by some defense methods.

#### 6.2.1. Trigger Stealthiness

As shown in Table 2, our attack method achieves higher PSNR values compared to the other two methods across three datasets. Specifically, the SSIM is higher than the other two attacks on the CIFAR-10 dataset. On the GTSRB and ISIC-2019 datasets, the SSIM values are comparable to the other two attacks. Therefore, the poisoned images generated by our method not only have the highest PSNR but also a higher SSIM value, making them difficult to distinguish from the original clean images. This indirectly demonstrates the increased stealthiness of the backdoor triggers generated by our approach.

As shown in Figure 6, the first column represents the original images, while the second, third, and fourth columns correspond to the pixel trigger, blend trigger, and frequency trigger, respectively. The second row displays the difference in images between the trigger-added images and the original images. It is observable that the images with the added triggers show almost no noticeable differences in appearance compared to the original images, making them hard to detect by the naked eye. Furthermore, in the residual image display, our method exhibits smaller pixel changes than the other two triggers. Therefore, the backdoor scheme based on frequency-domain triggers possesses excellent stealthiness, ensuring effective attack outcomes while avoiding detection.

#### 6.2.2. Attack Stealthiness (during the Training Phase)

In the context of backdoor attacks in federated learning, since malicious participants tend to produce updates with larger norms, a reasonable defense strategy is to ignore updates whose norms exceed a certain threshold Q. Assuming the adversary is aware of the threshold Q, we set this threshold to two. Granting the adversary this significant advantage makes the norm boundary defense effectively equivalent to the following method of norm clipping:

In the experiment, we set Q to 50, and the results are shown in Figure 7. Our method can finely control the magnitude of model updates during local training using the PGD method. Therefore, it can more effectively evade norm clipping defenses. This implies that our attack strategy can be subtly adjusted to fit within the constraints of the defense mechanism, thus maintaining its effectiveness while reducing the likelihood of detection.

Weak differential privacy involves adding Gaussian noise with a standard deviation of σ to the global model. In our experiment across the three datasets, the values of noise were set to 0.005, 0.001, and 0.002, respectively.

As shown in Figure 8, we recorded the values of ACC for the main task and ASR for the backdoor task for different Gaussian noise coefficient models, and we can find that the success rate of our backdoor attack is still high under the weak differential privacy setting, i.e., the noise is taken to be very small.

#### 6.2.3. Attack Stealthiness (After the Training Phase)

Firstly, we evaluated the stealthiness of our method using neural clean, a widely used pattern and optimization-based approach for mitigating backdoored models. Specifically, it involves searching for the optimal ‘poisoning’ pattern for each possible target label. The method then quantifies whether any optimal backdoor trigger pattern exists, using a metric known as the anomaly index. This index helps to identify deviations from normal model behavior. If the anomaly index for any class exceeds a threshold of two, the model is suspected of having a backdoor.

Fine pruning focuses more on the analysis of neurons. Given a specific layer of the neural network, this method analyzes the neuron responses to a set of clean images and detects dormant neurons. These dormant neurons are suspected of being associated with the backdoor. By identifying and pruning these dormant neurons, which are activated primarily or only by the backdoor triggers and not by normal inputs, the method aims to eliminate the backdoor from the model.

In the spatial domain, frequency manifests as global noise. Consequently, the triggers reverse-engineered by the neural clean method are relatively large, resulting in a low anomaly index and rendering them undetectable, which is shown in Table 3; during fine pruning, the success rate of our backdoor attack gradually decreases as the proportion of model pruning increases. However, as shown in Figure 9, the rate of decline in our method is smaller compared to baseline methods, indicating that our approach is more stealthy against pruning strategies.

### 6.3. Ablation Experiment

#### 6.3.1. The Impact of Trigger Hyperparameter α

The frequency trigger requires the fusion of spectral information from two images, and the hyperparameter alpha controls the contribution ratio of the trigger image’s information. Therefore, we first analyze the impact of this parameter on the effectiveness of the attack.

From Table 4, we can observe that the influence of alpha on the attack success rate is not significant. When the value of alpha is greater than 0.15, the backdoor can already be efficiently injected and triggered. However, an excessively high alpha value will make the fused image more closely resemble the trigger image, necessitating a careful selection of alpha. As alpha increases, the effectiveness of the attack improves, but its visual stealthiness decreases.

#### 6.3.2. The Impact of PGD Hyperparameter ε

During the backdoor injection process, we employ the PGD method to control the magnitude of model updates. Specifically, we use the hyperparameter *ε* to limit the L2 Norm of model updates. Consequently, we next analyze the impact of this parameter on the effectiveness of the attack. In our experiment, we record the attack success rate of the local models submitted by attackers under different values of *ε*. This analysis will help in understanding how the constraint on the update magnitude, imposed by *ε*, influences the ability of the attacker to successfully implement the backdoor without being detected.

Although a smaller ϵ value results in smaller updates to the backdoor model in each training round, making it more difficult to detect, it is observed from Figure 10 that when the model update magnitude is too low, it becomes more challenging for the attacker to inject the backdoor into the model. This leads to a decreased success rate of the backdoor attack on the global model, significantly impacting the efficiency of the attack.

## 7. Conclusions

In this paper, we propose a federated learning backdoor attack based on frequency-domain injection. First, the spectral magnitude of the two images is linearly mixed by Fourier transforming the trigger and the clean image, and the low-frequency information of the trigger is injected into the clean image, preserving the semantic information of the clean image. Second, the effectiveness of the attack is realized by expanding the weight update of the malicious client, and the PGD method is introduced on the server side to constrain the local model update and achieve the stealthiness of the attack. Experiments show that the attack success rate maintains good results in the single attack, continuous attack, and multi-trigger attack scenarios, while the trigger humans generated by our method are imperceptible to the naked eye and have high PSNR and SSIM values, which can attack common backdoor defense methods. In conclusion, the federated learning backdoor attack method based on frequency-domain injection has better attack effectiveness and stealthiness.

Inspired by image steganography methods, future work may combine our method with image steganography methods to realize frequency-domain-based image steganography in federated learning backdoor attack scenarios, which is important for intellectual property protection. Also, other frequency-domain injection methods, such as a discrete Fourier transform, are considered for application in federated learning backdoor attack scenarios.

## Figures and Tables

**Figure 1 entropy-26-00164-f001:**
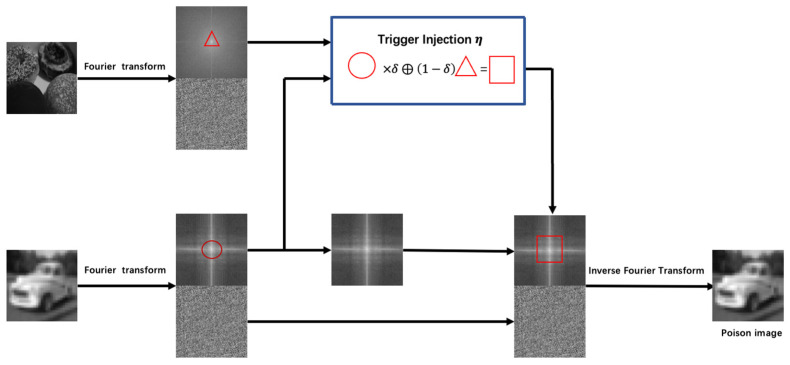
The frequency-domain poisoning sample generation process.

**Figure 2 entropy-26-00164-f002:**
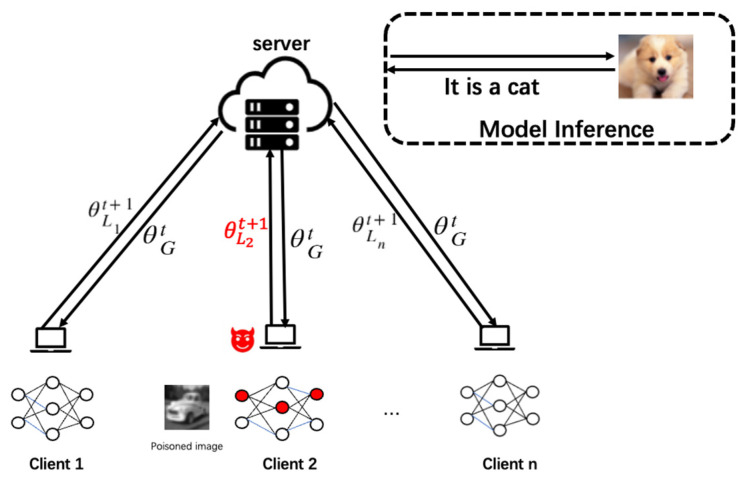
The federated learning backdoor submission process.

**Figure 3 entropy-26-00164-f003:**
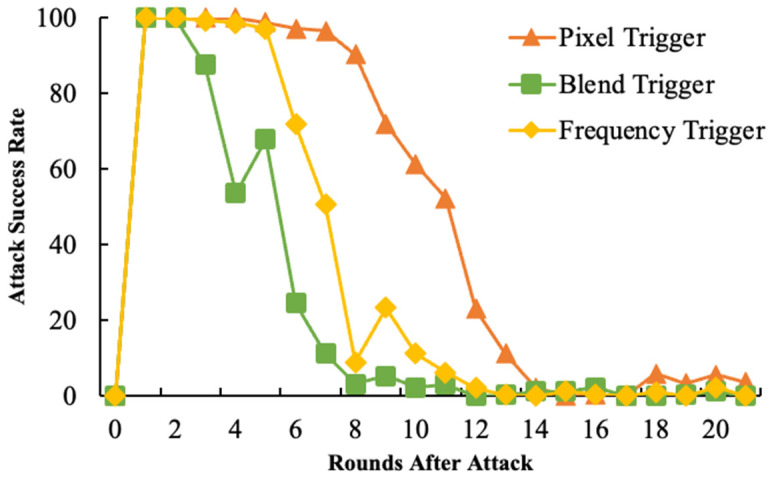
The single attack success rate.

**Figure 4 entropy-26-00164-f004:**
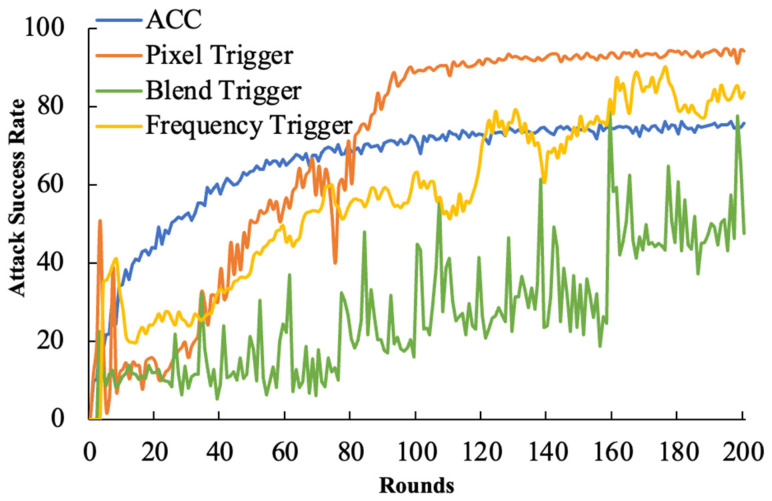
The continuous attack success rate.

**Figure 5 entropy-26-00164-f005:**
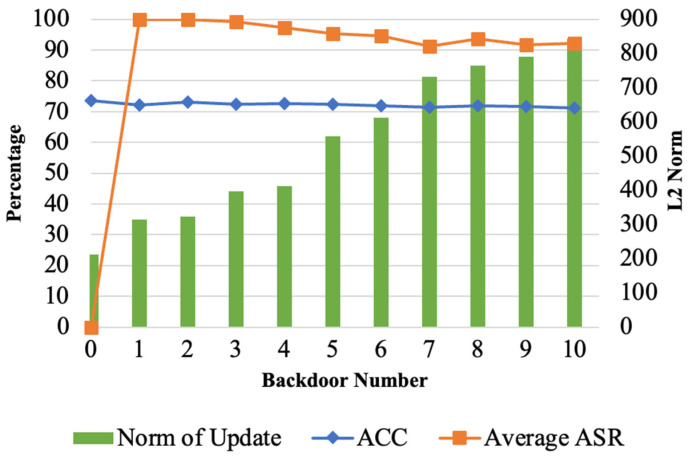
Multi-backdoor attack assessment.

**Figure 6 entropy-26-00164-f006:**
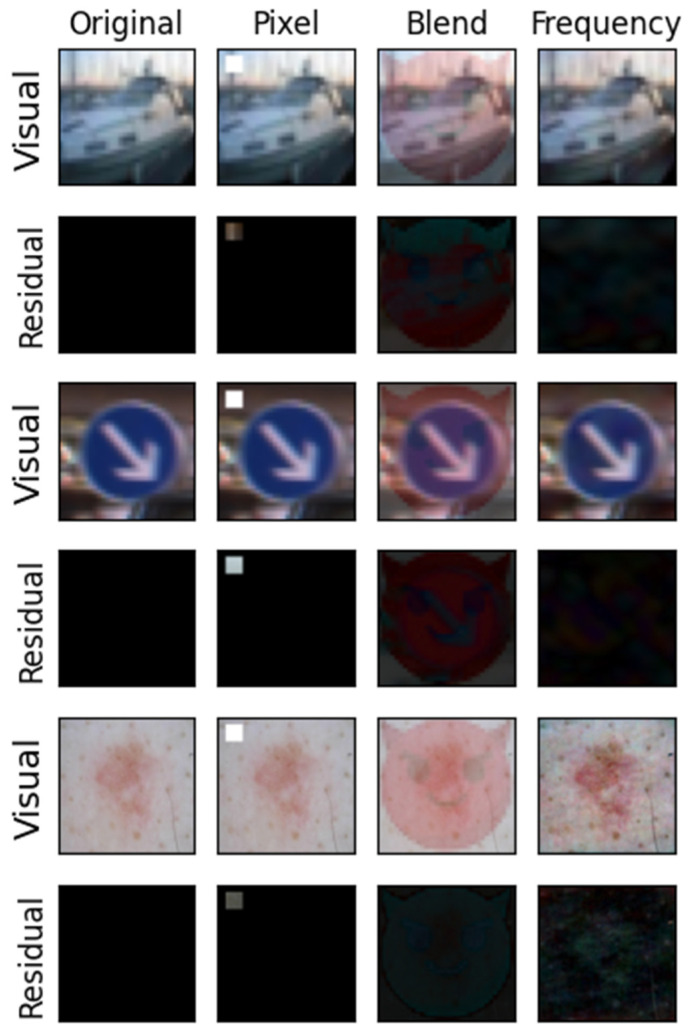
The backdoor trigger display.

**Figure 7 entropy-26-00164-f007:**
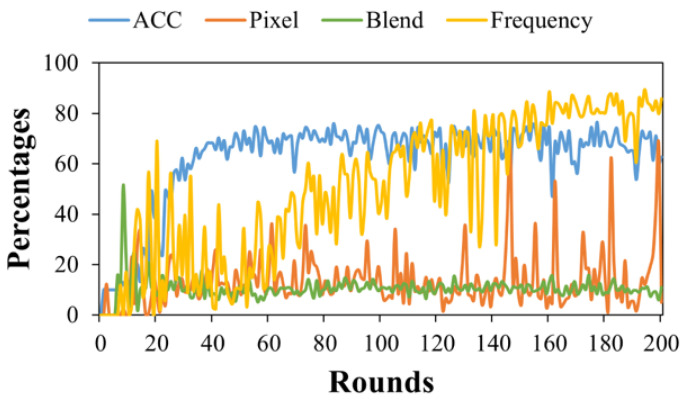
The norm clipping defense.

**Figure 8 entropy-26-00164-f008:**
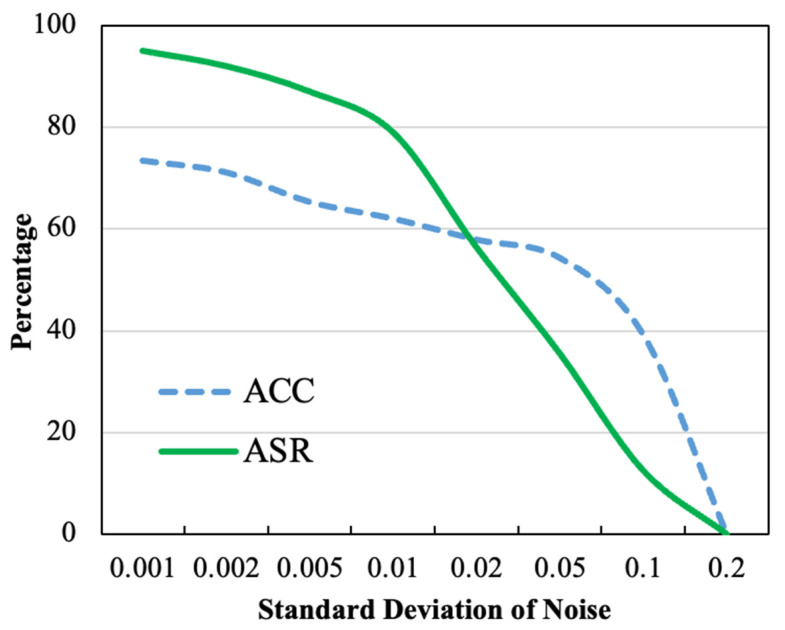
The weak differential privacy defense.

**Figure 9 entropy-26-00164-f009:**
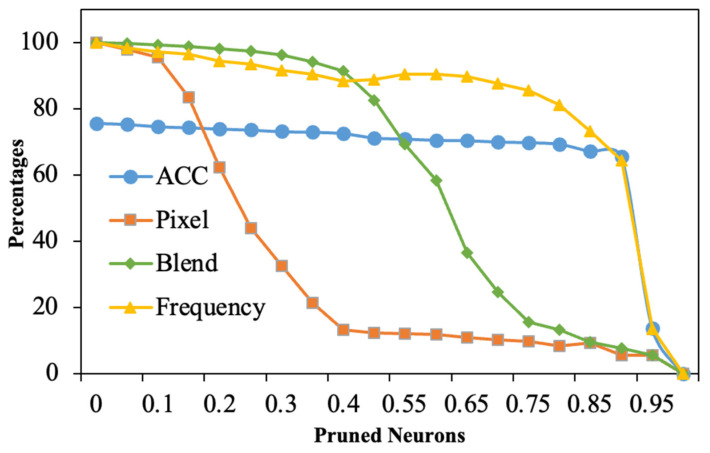
The fine-pruning defense.

**Figure 10 entropy-26-00164-f010:**
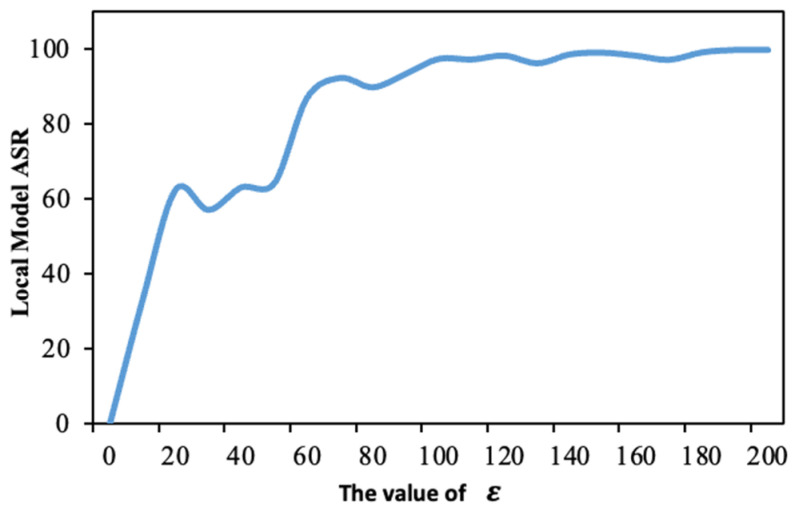
The attack success rate of the model submitted by the attacker under different *ε*.

**Table 1 entropy-26-00164-t001:** ASR and ACC results under different datasets and different models.

Model	Attack Scheme	Datasets	Benign Acc	iid		Non-Lid	
				ASR	ACC	ASR	ACC
ResNet18	Piexl pattern	CIFAR-10	75.66	99.97	74.97	9994	72.68
		GTSRB	92.93	99.93	91.87	99.98	89.87
		ISIC-2019	77.79	99.62	78.24	99.37	74.34
	Blend	CIFAR-10	75.66	98.98	74.87	99.42	71.98
		GTSRB	92.93	98.74	92.13	99.34	91.34
		ISIC-2019	77.79	97.95	79.23	98.73	75.64
	Frequency (ours)	CIFAR-10	75.66	99.34	75.79	99.64	72.35
		GTSRB	92.93	99.67	92.32	99.57	91.36
		ISIC-2019	77.79	99.23	78.02	99.34	75.24
VGG13	Piexl pattern	CIFAR-10	73.58	99.84	73.24	98.67	70.87
		GTSRB	91.23	99.68	91.43	99.35	88.64
		ISIC-2019	75.34	99.42	75.39	99.71	73.14
	Blend	CIFAR-10	73.58	99.37	74.27	98.14	72.19
		GTSRB	91.23	98.93	91.23	99.93	88.24
		ISIC-2019	75.34	98.84	77.32	98.64	74.36
	Frequency (ours)	CIFAR-10	73.58	99.76	73.21	99.23	71.23
		GTSRB	91.23	99.86	92.08	99.57	87.35
		ISIC-2019	75.34	99.72	76.52	99.83	74.86

**Table 2 entropy-26-00164-t002:** SSIM and PSNR values under different datasets.

Attack Methods	CIFAR10		GTSRB		ISIC-2019	
	PSNR	SSIM	PSNR	SSIM	PSNR	SSIM
Pixel pattern	25.99	0.968	22.30	0.960	25.11	0.987
Blend	23.9	0.929	22.49	0.872	25.59	0.909
Frequency	29.95	0.977	29.56	0.944	26.58	0.892

**Table 3 entropy-26-00164-t003:** The neural clean defense.

Model	Clean	Pixel	Blend	Frequency
Anomaly Index	0.94	3.12	1.72	1.38

**Table 4 entropy-26-00164-t004:** The attack success rate under different α values.

α	ASR
0.05	92.36
0.10	96.24
0.15	99.32
0.20	99.36
0.25	98.45
0.30	99.36

## Data Availability

Data are available on request from authors.

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
