# Peer review of "Federated Learning Backdoor Attack Based on Frequency Domain Injection"

_entropy, 2024, doi:10.3390/e26020164_

Round 1
Reviewer 1 Report
Comments and Suggestions for Authors
The manuscript suggests a technique where a backdoor attack is conducted by manipulating the frequency domain of images in the context of machine learning or image processing. The objective is to manipulate the clean image by injecting the low-frequency information of the trigger while attempting to preserve the semantic information of the clean image. In the context of machine learning, this approach could be used as a sophisticated form of a backdoor attack. Defenses against such attacks may involve robust preprocessing techniques, anomaly detection during training, or methods to detect and neutralize the injected triggers.
The content of the manuscript is interesting from a research perspective. The manuscript is mostly clearly and interestingly written. The only problem is with formatting and alignment of equations, figures and tables. Also with some of the headings and text being a different colour shade to the rest of the font (6.1.2. One-shot attack and Continuous Attack). I would recommend correcting the formatting of the manuscript. Also, section "6.1 Backdoor attack effectiveness, 6.2. Backdoor attack stealthiness" is inappropriately numbered. There is no text in it. Incorrect formatting of the algorithm 4. Method and its description. Lack of caption of algorithms as is the case with the figures. The same problem with the text color.
From the scientific content of the article, the algorithm presented (Algorithm 1) is not fully clear in terms of some steps. Adding more detailed explanations, especially for complex operations like the Fourier Transform, can enhance the clarity for readers. The ASR and ACC are commonly used metrics, providing additional information on why these metrics were chosen and how they align with the goals of the attack would strengthen the evaluation section. Consider discussing the relevance of SSIM and PSNR in the context of backdoor attacks. A study on the impact of trigger hyperparameters and PGD hyperparameters is a valuable addition. However, a more thorough discussion of the implications of these results on the overall attack strategy could enhance the paper.
The conclusion is not very elaborated considering the content and results. It also lacks further directions for future research.
I would recommend:
1. to revise the formatting according to the comments, MDPI template
2. to update the conclusion section according to the comments
3. to address the scientific points
Author Response
The manuscript suggests a technique where a backdoor attack is conducted by manipulating the frequency domain of images in the context of machine learning or image processing. The objective is to manipulate the clean image by injecting the low-frequency information of the trigger while attempting to preserve the semantic information of the clean image. In the context of machine learning, this approach could be used as a sophisticated form of a backdoor attack. Defenses against such attacks may involve robust preprocessing techniques, anomaly detection during training, or methods to detect and neutralize the injected triggers.
The content of the manuscript is interesting from a research perspective. The manuscript is mostly clearly and interestingly written.
Response: We thank the reviewer for the positive evaluation of our work and we present our point-to-point responses to the comments in the following.
Comment 1:. The only problem is with formatting and alignment of equations, figures and tables. Also with some of the headings and text being a different colour shade to the rest of the font (6.1.2. One-shot attack and Continuous Attack). I would recommend correcting the formatting of the manuscript.
Response: Thank you for your careful and professional comments .We have carefully revised the formatting and alignment of formulas, figures and tables. The color shades of some headings and text were also corrected. And the MDPI template format was strictly adhered to.
Comment 2: Also, section "6.1 Backdoor attack effectiveness, 6.2. Backdoor attack stealthiness" is inappropriately numbered. There is no text in it. Incorrect formatting of the algorithm 4. Method and its description. Lack of caption of algorithms as is the case with the figures. The same problem with the text color.
Response: Thank you for your careful and professional comments .We renumbered the headings and carefully numbered the formulas, tables, and figures. Overview text was added below the 6.1 Backdoor attack effectiveness, 6.2. Backdoor attack stealthiness sections. Figure 2 was described and modified as follows:
Added summary text:
Below part 6.1:
In this section, we conduct experiments on the effectiveness of federated learning backdoor attacks. The ASR and ACC on two models for three datasets are evaluated, as well as the effectiveness of the attack in One-shot Attack, Continuous attack and Multiple Trigger.
Below part 6.2:
In this section, we conducted experiments on the stealthiness of federated learning backdoor attacks. The stealthiness of the poisoned images was measured by visualizing the residual images and calculating PSNR and SSIM metrics. The stealthiness and effectiveness of our method is also evaluated by some defense methods.
A description of Figure 2:
As shown in Fig. 2, the attacker adds the poisoned samples generated by the frequency domain based injection to the local client 2 and implants a backdoor to the global model by augmenting the parameters of the malicious client.When the global model with the backdoor is tested, the photo with the trigger dog is recognized as a cat.
Comment 3: From the scientific content of the article, the algorithm presented (Algorithm 1) is not fully clear in terms of some steps. Adding more detailed explanations, especially for complex operations like the Fourier Transform, can enhance the clarity for readers.
Response: Thank you for your careful and professional comments. Our specific modifications are as follows:
|
Algorithm 1 Federated learning backdoor attack based on frequency injection |
|
Input: benign dataset, benign image , trigger image, Local batch size , number of local training round R, the global model after the t th round of aggregation, the latest local model of client C after round t+1 Output: Malicious client model update Stage1: Create poisoned image 1. for all do: 2. Performing Faset Fourier Transform of and // Perform Eq. 4 to perform a fast Fourier transform 3. // Eq. 5 is executed to obtain and , and the amplitude spectra of the trigger image and the clean image are mixed by a binary mask matrix . 4. // is obtained through Eq. 5, and the original phase and amplitude spectra are synthesized for the inverse Fourier transform to get 5. add to get the 6. end for 7. return to poisoned training set Stage2: Federated learning backdoor attack 8. The server sends global model parameters to client and updates the local model 9. for R=1,…R do: 10. (split into batches of size B) 11. for do: 12. // Perform stochastic gradient descent algorithm to update the local model 13. if do:// Execute the PGD algorithm to constrain the local model update magnitude not to exceed a given threshold value 14. // denotes the L2 norm of the update weights 15. end for 16. end for 17. return to server |
Comment 4: The ASR and ACC are commonly used metrics, providing additional information on why these metrics were chosen and how they align with the goals of the attack would strengthen the evaluation section.
Response: Thank you for your careful and professional comments. Our specific modifications are as follows:
ASR:Attack Success Rate, which is the rate of backdoor samples that are successfully classified as target labels. Where ASR is used to measure the recognition accuracy of the backdoor model for backdoor data, where accuracy refers to the ability to accurately recognize the backdoor image as the target label, rather than its true label. The closer to 1 means the stronger the attack.
ACC: The model's classification success rate on clean samples, i.e., the ratio of the number of samples correctly predicted by the model to the total number of samples. ACC is used to measure the recognition accuracy of the backdoor model on clean data, where recognition accuracy refers to the ability to accurately recognize clean samples as true labels. The closer to 1 indicates the better performance of the model.
Comment 5: Consider discussing the relevance of SSIM and PSNR in the context of backdoor attacks. A study on the impact of trigger hyperparameters and PGD hyperparameters is a valuable addition. However, a more thorough discussion of the implications of these results on the overall attack strategy could enhance the paper.
Response: Thank you for your careful and professional comments. Our specific modifications are as follows:
PSNR (Peak Signal-to-Noise Ratio): As the name suggests, PSNR measures the pixel error corresponding to the addition of the frequency-domain trigger poisoning image and the original clean image . A larger PSNR implies less distortion in the generated poisoned image.
PSNR is defined as:
Where indicates the maximum value of the image color.
SSIM (Structural Similarity Index):.SSIM is a metric used to measure the similarity of two images. It is calculated based on the brightness and contrast of local patterns. For SSIM, the value ranges from 0 to 1. The higher the finger of similarity index (almost close to 1) means that the poisoned image is closer to the original clean image. Defined as:
Where, and are poisoned and clean samples, respectively, and are sample pixel mean values; and are sample pixel variance values. is the sample pixel covariance; and the constants to maintain the stability, , is the pixel value dynamic range. L is the dynamic range of pixel values.
If the attacker executes the Stochastic Gradient Descent(SGD) algorithm for too long, then the generated model can deviate severely from its origin, thus making a simple paradigm tailoring defense effective. Therefore, in order to improve the stealthiness of backdoor attacks, we further propose to use PGD to constrain the update of the local model from exceeding a given threshold ϵ during backdoor injection by the local client:
The adversary then runs PGD where the projection happens on the ball centered around with radius
Comment 6: The conclusion is not very elaborated considering the content and results. It also lacks further directions for future research.
Response: Thank you for your careful and professional comments. Our specific modifications are as follows:
In this paper, we propose a federated learning backdoor attack based on frequency domain injection. First, the spectral magnitude of the two images is linearly mixed by Fourier transforming the trigger and the clean image, and the low-frequency information of the trigger is injected into the clean image, preserving the semantic information of the clean image. Second, the effectiveness of the attack is realized by expanding the weight update of the malicious client, and the PGD method is introduced at the server side to constrain the local model update and achieve the stealthiness of the attack. Experiments show that the attack success rate maintains good results in single attack, continuous attack and multi-trigger attack scenarios, while the trigger humans generated by our method are imperceptible to the naked eye and have high PSNR and SSIM values, which can attack common backdoor defense methods. In conclusion, the federated learning backdoor attack method based on frequency domain injection has better attack effectiveness and stealthiness.
Inspired by image steganography methods, future work may combine our method with image steganography methods to realize frequency-domain based image steganography in federated learning backdoor attack scenarios, which is important for intellectual property protection. Also, other frequency domain injection methods, such as discrete Fourier transform, are considered for application in federated learning backdoor attack scenarios.

Reviewer 2 Report
Comments and Suggestions for Authors
In FL models, a few malicious nodes should have a minor effect on the global model. I would be interested to know what the minimum percentage of malicious nodes is, which could have a significant effect on the global model.
Author Response
Comment :In FL models, a few malicious nodes should have a minor effect on the global model. I would be interested to know what the minimum percentage of malicious nodes is, which could have a significant effect on the global model.
Response: Thank you for your careful and professional comments. In our paper, one-shot experiments are conducted to inject backdoor into the global model when there is a single malicious node, which affects the global model when the minimum percentage of malicious nodes is 0.5% to 1%. When the percentage of malicious nodes is small, the convergence of the model is slow, but as long as the runtime is long, the backdoor will be successfully injected into the global model.
